# Does environmental regulation reduce China's haze pollution? An empirical analysis based on panel quantile regression

**Congxin Li**[ॐ], **Guozhu Li**[iD][ॐ]*

School of Economics, Hebei GEO University, Shijiazhuang, Hebei Province, China

[ॐ] These authors contributed equally to this work.
* guozhuli@126.com

**Data Availability Statement:** All relevant data are within the manuscript and its Supporting Information files.

## Abstract

Haze pollution in China is very serious and has become the source of mortality, affecting the health and lives of residents. The Chinese government adopts different policy measures to reduce haze pollution. The impact of different types of environmental regulations on haze pollution has become a hot topic for academics and government departments. Based on panel data from 2005–2017, this paper studies the effect of different types of environmental regulations on haze pollution in 30 provinces of China using a panel quantile model. The results show that when haze pollution changes from a low quantile to a high quantile, the marginal impact of command-and-control environmental regulation on haze pollution changes from 0.122 to -0.332. Command-and-control environmental regulation can reduce haze pollution, but its impact is not significant. The main reason for this finding is that environmental law enforcement is not strict. The marginal impact of economically restrained environmental regulation on haze pollution changes from -14.389 to 49.939. Economically restrained environmental regulation can reduce haze pollution in low quantiles, but not in high quantiles. The collection of sewage charges fees is far less than the total profit, which has no deterrent effect on enterprises. The marginal impact of public participation in environmental regulation on haze pollution changes from 0.154 to -0.002. Public participation in environmental regulation cannot reduce haze pollution in low quantiles, but can in high quantiles; however its impact becomes insignificant. This study reveals the quantile-based discrepancy in the effect of environmental regulation on haze pollution, and offers a new perspective for research on the effects of environmental regulation.

## Introduction

While achieving great accomplishments, the growth of the Chinese economy has led to a series of ecological problems, among which haze pollution is the most serious. In 2013, China suffered the worst hazy weather in history, affecting 25 provinces and more than 100 large and medium-sized cities. Haze pollution has a negative impact on travel and the health of Chinese [1–4]. ***The Air Pollution Prevention and Control Action Plan*** issued by the government in

**Funding:** This research was supported by the Projects of Chinese National Funding of Social Sciences (No.18BJY081).

**Competing interests:** All authors declare that there are no conflicts of interest in this paper.

September 2013 is the most stringent atmospheric governance action plan in China's history. Since then, the central government has promulgated a number of rules and regulations for air pollution control [5], and local governments at all levels have reached a broad consensus to break regional administrative boundaries [6] and coordinate local haze prevention and control policies [7, 8] to jointly control air pollution.

In addition to meteorological factors [9, 10], China's economic development mode [11, 12], industrial structure [13, 14], and energy-use efficiency [15, 16] lead to the frequent occurrence of hazy weather. Jin et al. [17] pointed out that fine particulate matter ($PM_{2.5}$) in various provinces in China increased to different degrees from 2005 to 2014. China's governments at all levels have taken various measures, and China's haze pollution level dropped by 9.36% annually from 2013 to 2016 [18]. However, *the 2018 Global Environmental Performance Index (EPI) report* released by Yale University in the United States pointed out that China's air quality performance is poor, ranking 177[th] among 180 countries [19]. Improving air quality is still the top priority for China's economic and social development.

General secretary Xi Jinping pointed out that only by implementing the strictest system and the strictest rule of law can we provide reliable guarantees for the construction of ecological civilizations. Without institutional guarantees, the concept of green development is a castle in the air, and the management of haze pollution will be difficult to sustain [20–22]. In this paper, a panel quantile regression model is used to study the degree of influence of environmental regulation on haze pollution at different quantile levels. Compared with a general mean regression, the panel quantile regression takes into account the heterogeneity of the impact and the results are more robust.

## Literature review

The research on haze pollution mainly focuses on three aspects.

Research on influencing factors of haze pollution. Some scholars focus on economic development and believe that economic development leads to the aggravation of haze pollution [23–25]. Some scholars believe that the urbanization process [26–28], industrial structure [29–31], energy consumption structure [32–34] and foreign investment [35–38] are also important factors leading to the increase in haze pollution. Although these scholars focus on a certain angle, a number of control variables, such as technology investment, transportation, greening level, population density, and human capital, have been added to the actual research. In addition to the core explanatory variables, these control variables also have different effects on haze pollution. The above scholars study the influencing factors that cause haze pollution. This paper focuses on whether haze pollution can be reduced through economically restrained environmental regulation.

Research on the negative effect of haze pollution. The first negative effect of haze pollution is harm to the human body. Othman et al. [39] found that hospitalizations increased by 31% on hazy days compared to normal days. Haze pollution can cause respiratory diseases, such as asthma, bronchitis and emphysema [40–42], and has a major impact on the human cardiovascular system, nervous system and immune system [43]. The second negative effect of haze pollution is harm to traffic and the living environment, which indirectly leads to very large economic losses. Through questionnaire survey, Zhang found that 91% of respondents believed that haze had a great impact on tourism traffic (especially aviation) [44]. Environmental pollution, represented by hazy weather, is one of the important reasons for the decline in the number of tourists [45–47]. In the hazy weather, people's concerns about travel safety and health risks will lead to cancellations of their original travel plans, which will have some impact on the Chinese economy. These scholars analyze the negative impact of haze pollution on

public health and life. When the public perceives the adverse effects of haze pollution, can they effectively participate in the control of haze pollution? The purpose of our study is to determine whether public participation in environmental regulation can reduce haze pollution.

Research on haze pollution governance. Haze pollution has become a roadblock on the road to the construction of ecological civilization, which has led the Chinese government and all sectors of society to actively explore effective ways to control haze pollution. Some scholars advocate the use of environmental policies to reduce haze pollution [48–51]. Because haze pollution has no boundary, China should ignore provincial boundaries, jointly defend and control resources, and cooperate to govern haze pollution [52]. In addition, the intrinsic mechanism and prevention of haze pollution [53–55], as well as foreign experience in haze treatment [56, 57], all provide ideas for the management of haze pollution. These scholars have proposed various measures in the management of haze pollution. China's top leader also promotes the strictest rule of law to tackle environmental pollution. Whether command-and-control environmental regulation can reduce haze pollution is one of the topics studied in this paper.

In summary, different scholars have studied haze pollution from different angles, providing good references and inspiration for this research. Through the collation of information regarding the impact of environmental regulation on haze pollution, it is found that the influence of environmental regulation on haze pollution has led to different conclusions due to different indicators adopted by different scholars'. One conclusion is that environmental regulation does not alleviate haze pollution [58, 59]. Another conclusion is that environmental regulation has a certain inhibitory effect on haze pollution [60, 61]. Whatever the conclusion, the econometric models adopted by these scholars for empirical testing are mean regressions. These scholars analyze the effect of the explanatory variables on the average level of explained variables. (i.e., impact of environmental regulation on the average haze level). Previous analyses do not reflect the impact of environmental regulation on changes under different levels of haze pollution. This paper studies the variation of the coefficient in environmental regulation on different haze pollution levels, and explores whether the relationship between variables will change.

## Methods and data source

### Methods

Classical regression models, which are actually mean regressions, focus on the influence of an explanatory variable ($x$) on the conditional expectation ($E(y|x)$) of an explained variable (y). If the conditional distribution ($y|x$) is not symmetrical, the conditional expectation ($E(y|x$) cannot easily reflect the full picture of the entire conditional distribution. Koenker and Bassett proposed quantile regression [62], which provides a more comprehensive understanding of conditional distributions ($y|x$) by estimating several important conditional quantiles, such as $25^{th}$ quantile, median, and $75^{th}$ quantile.

Assuming that the overall quantile ($y_q(x)$) of the conditional distribution ($y|x$) is a linear function of $x$, the quantile regression model is:

$$y_q(\mathbf{x_i}) = \mathbf{X_i'}\boldsymbol{\beta_q} + u_i \quad (i = 1, 2, \ldots, n) \tag{1}$$

where $0 < q < 1$ is the quantile, $\boldsymbol{\beta_q}$ denotes the regression coefficient of quantile $q$, and the estimator can be obtained by Formula 2.

$$\hat{\boldsymbol{\beta}}_{\mathbf{q}} = \arg\min_{\boldsymbol{\beta_q}}(\sum_i q|y_i - \mathbf{X_i'}\boldsymbol{\beta_q}| + \sum_i (1-q)|y_i - \mathbf{X_i'}\boldsymbol{\beta}|) \tag{2}$$

Quantile regression is applied to panel data to construct a panel quantile regression model.

$$y_q(\mathbf{x_{it}}) = \alpha_i + \mathbf{X'_{it}}\boldsymbol{\beta_q} + u_{it} \quad (i = 1, 2, \ldots, n; t = 1, 2, \ldots, T) \tag{3}$$

where $\alpha_i$ denotes individual heterogeneity that does not change over time and $u_{it}$ indicates a random error term.

According to Formula 3, the parameter estimator of the panel quantile regression model can be expressed as,

$$(\hat{\alpha}_q, \hat{\boldsymbol{\beta}}_q) = \underset{\hat{\alpha}_q, \hat{\beta}_q}{argmin}\{\sum_i\sum_t \rho_q(y_{it} - \alpha_{i(q)} - \boldsymbol{x'_{it}}\boldsymbol{\beta_q}) + \lambda\sum_i|\alpha_{i(q)}|\} \tag{4}$$

where $\hat{\alpha}_q, \hat{\boldsymbol{\beta}}_\mathbf{q}$ is the penalty quantile regression estimator when $\lambda > 0$, and $\hat{\alpha}_q, \hat{\boldsymbol{\beta}}_\mathbf{q}$ a is the fixed effect quantile estimator when $\lambda = 0$.

## Variable descriptions

**Explained variable.** The explained variable is haze pollution ($Hap_{it}$). The main culprit for aggravating haze pollution is respirable particles, the main components of which include $PM_{2.5}$ and $PM_{10}$. $PM_{2.5}$ is a particle with a diameter of 2.5 microns or less, which damages residents' living and atmospheric environments more than $PM_{10}$. Therefore, this paper uses the average annual concentration of $PM_{2.5}$ (micrograms/cubic meter) in different provinces to measure haze pollution.

**Core explanatory variables.** The core explanatory variable is environmental regulation. Li and Liu used a fixed efficiency model and found that environmental regulation had a significant effect on promoting green economic efficiency for a long time [63]. Environmental regulation is conducive to the green development of China's economy. There are many indicators for environmental regulation. With reference to Yang and Hu [64], this paper constructs three measurement indicators: command-and-control environmental regulation ($Cor_{it}$), economically restrained environmental regulation ($Ecr_{it}$), and public participation in environmental regulation ($Pur_{it}$). Command-and-control environmental regulation is calculated by dividing the number of environmentally-related administrative penalties (pieces) by the number of industrial enterprises above a designated size. The larger the number, the stronger the environmental regulation. Economically restrained environmental regulation is calculated by dividing the sewage fee amount (100 million yuan) by the total profit of industrial enterprises above a designated size (100 million yuan). The greater the proportion of sewage fees to total profits, the stronger the environmental regulations. Public participation environmental regulation is calculated by dividing the number of environmental pollution petitions (times) by the number of industrial enterprises above a designated size. The larger the value, the stronger the environmental regulation.

**Control variables.** Control variables include industrial concentration ($Inc_{it}$), enterprise scale ($Ens_{it}$) and science and technology input intensity ($Sci_{it}$). The degree of industrial concentration is calculated by dividing the total assets of the industrial enterprises above a designated size (100 million yuan) by the number of industrial enterprises above a designated size. The higher the degree of industrial concentration, the more serious the haze pollution. The enterprise scale is calculated by dividing the main business income of industrial enterprises above the scale by the number of industrial enterprises above the scale (RMB 100 million). The larger the enterprise, the more capable it is to carry out technological innovation and thus improve the environmental quality. The science and technology input intensity can be calculated by dividing the research and development (R&D) investment (100 million yuan) by the regional gross domestic product (GDP,100 million yuan). Research and development (R&D) refers to

systematic and creative activities in the field of science and technology aimed at increasing knowledge and the use of knowledge for new applications. The efficiency of science and technological services is conducive to pollution control [65]. Gross domestic product (GDP) refers to the final products produced by all residential units in a country during a certain period of time. If science and technology are put into use for clean environmental protection technology innovation, haze pollution will be reduced.

According to the theoretical model and variable descriptions, the panel quantile model of the impact of environmental regulation on haze pollution is constructed.

$$
\begin{aligned}
Hap_{it}(q|\boldsymbol{x_{it}}) = &\, \alpha_i(q) + \beta_1(q)Cor_{it} + \beta_2(q)Ecr_{it} + \beta_3(q)Pur_{it} + \beta_4(q)Inc_{it} \\
&+ \beta_5(q)Ens_{it} + \beta_6(q)Sci_{it} + u_{it} \quad (i = 1, 2, \ldots, n; t = 1, 2, \ldots, T)
\end{aligned}
\tag{5}
$$

where $i$ and $t$ indicate the province and time, $q$ indicates the quantile, $0 < q < 1$, $\beta_k(q)$ denotes the regression coefficient of quantile $q$, which varies with the change in $q$. The other variables are the same as above.

## Data source

The research object of this paper is China's 30 provinces (excluding Tibet, Hong Kong, Macao and Taiwan) from 2005 to 2017. In February 2012, the Ministry of Environmental Protection of China officially issued a new *Environmental Air Quality Standard*. By the end of the year, some cities began to collect data related to $PM_{2.5}$. Therefore, $PM_{2.5}$ from 2005–2016 comes from the global satellite grid data released by Columbia University. Because these data are basically consistent with the Chinese Ministry of Environmental Protection's judgment on the haze situation, the $PM_{2.5}$ for 2017 was collected from the *China Environment Yearbook* (2018). The sewage fee amounts and the number of petitions related to environmental pollution came from the *China Environment Yearbook* (2006–2018). The total profit of industrial enterprises above the designated size, the total assets of the industrial enterprises above the designated size, the main business income of the industrial enterprises above the designated size, the R&D investment, the regional GDP and the number of the industrial enterprises above designated size are from the *China Statistical Yearbook* (2006–2018.) The software used in the empirical analysis is Stata14.0.

## Empirical analysis of the impact of environmental regulations on haze pollution

### Description of the current situation of haze pollution in China

Fig 1 shows that some provinces have abnormal haze values from 2005 to 2017. The abnormal values appear on the right side of the box-line diagram, indicating that the provinces have the highest levels of haze pollution in these years. The abnormal values appear on the left side of the box-line diagram, indicating that the provinces have the lowest levels of haze pollution during these years. The average annual $PM_{2.5}$ concentration in 30 provinces of China from 2005 to 2017 was 41.3 micrograms per cubic meter, which was higher than the loose standard of 35 micrograms per cubic meter published by the World Health Organization (WHO) in 2012. From 2005 to 2017, the average annual concentration of $PM_{2.5}$ exceeded 41.3 micrograms per cubic meter in the provinces of Anhui, Beijing, Hebei, Henan, Hubei, Jiangsu, Shandong, Tianjin and Xinjiang. There are abnormalities in Chongqing (2 years) and Hunan (1 year) during which the $PM_{2.5}$ concentration was less than 41.3, but the remaining years have values greater than 41.3. Haze pollution less than 35 micrograms per cubic meter in all years occurred only in Fujian, Hainan, Heilongjiang, Neimenggu, Sichuan and Yunnan. The data show that

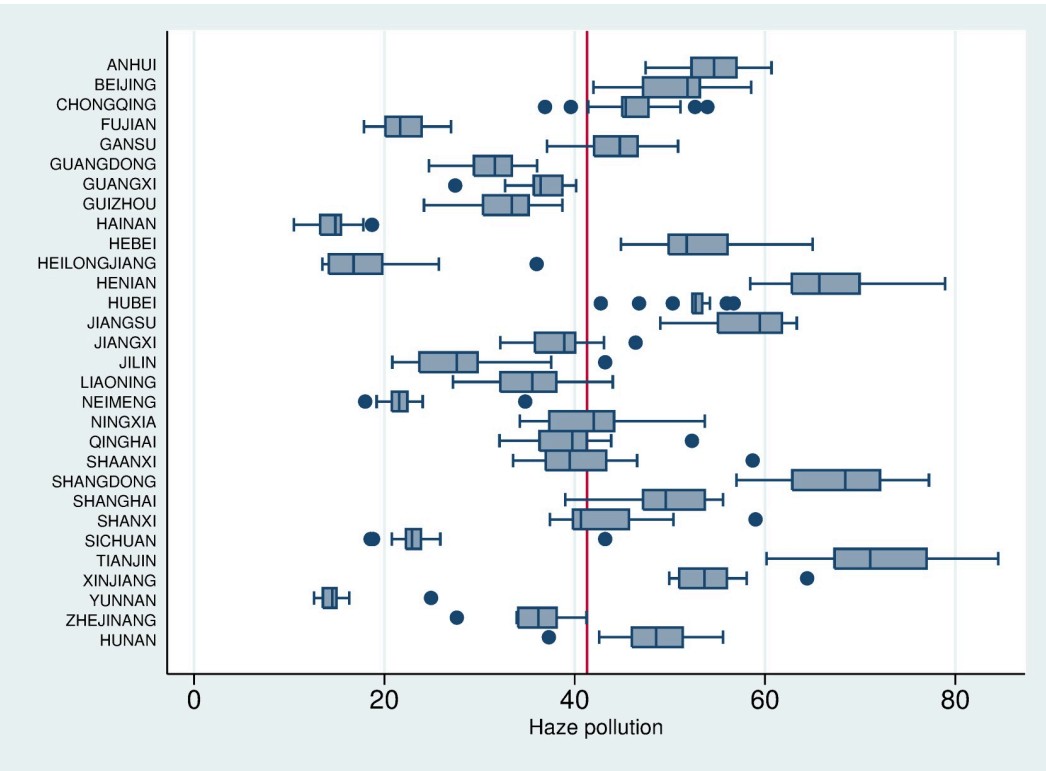

**Fig 1. Box plot of haze pollution status in 30 provinces of China from 2005 to 2017.**

the three provinces with the most severe haze pollution in 2005–2017 were Henan, Shandong and Tianjin.

## Analysis of the model results for the impact of environmental regulation on haze pollution

Quantile regression uses the weighted average of the absolute residual value of a minimized objective function, which is not easily affected by extreme values. Compared with the traditional panel regression model, the results are more robust. According to Formula 5, five representative quantile loci are selected in this paper:10%, 25%, 50%, 75% and 90%. The panel quantile regression results of the impact of environmental regulation on haze pollution are shown in Table 1.

Table 1 shows that the influence coefficient of the command-and-control environmental regulation on haze pollution changes from positive to negative. In the low quantile, the marginal effect of the command-and-control environmental regulation on haze pollution is 0.122. This indicates that command-and-control environmental regulation increases haze pollution. When haze pollution reaches an average level, the marginal effect of command-and-control environmental regulation on haze pollution is -0.071. Command-and-control environmental regulation may reduce haze pollution. In the high quantile, the marginal effect of the command-and-control environmental regulation on haze pollution is -0.322. The quantile coefficient of command-and-control environmental regulation is still negative, but the value becomes larger, indicating that the constraint of command-and-control environmental regulation on haze pollution is further strengthened. Command-and-control environmental

Table 1. Panel quantile regression results.

| Variable | Quantile | | | | |
|---|---|---|---|---|---|
| | q = 10% | q = 25% | q = 50% | q = 75% | q = 90% |
| Cor | 0.122 (0.834) | 0.027 (0.949) | -0.071 (0.835) | -0.209 (0.657) | -0.332 (0.639) |
| Ecr | -14.389 (0.266) | -0.904 (0.922) | 12.985* (0.090) | 32.484*** (0.002) | 49.939*** (0.001) |
| Pur | 0.154 (0.116) | 0.121* (0.085) | 0.088 (0.128) | 0.041 (0.609) | -0.002 (0.990) |
| Inc | -0.158 (0.811) | 0.242 (0.608) | 0.655* (0.093) | 1.235** (0.021) | 1.753** (0.028) |
| Ens | -1.193 (0.162) | -1.331** (0.030) | -1.472 *** (0.003) | -1.671** (0.016) | -1.850* (0.074) |
| Sci | -0.498 (0.789) | -0.622 (0.643) | -0.751 (0.494) | -0.931 (0.537) | -1.093 (0.630) |
| Obs | 390 | 390 | 390 | 390 | 390 |

Note:

* indicates significance at the 10% level,

** indicates significance at the 5% level,

*** indicates significance at the 1% level.

regulation can reduce haze pollution. However, the impact of command-and-control environmental regulation on haze pollution is not significant. The main reason for this finding is that environmental law enforcement is not strict, and there are laws that do not comply with the regulations. Some projects start construction without performing environmental assessments. When environmental pollution is created, warnings or fines will imposed, but the cost of violating the law is very low. There are certain difficulties associated with the investigation of environmental violations. After the investigation, the current environmental law standards have no corresponding deterrent effect on entities performing illegal actions.

The coefficient of influence of the economically restrained environmental regulation on haze pollution changes from negative to positive. In the low quantile, the marginal effect of the economically restrained environmental regulation on haze pollution is -14.389. The P value is 26.6%, which is greater than 5%. Economically restrained environmental regulations reduce haze pollution, but the effect is not significant. When haze pollution is in the 50th, 75th and 90th quantiles, the impact of economically restrained environmental regulation is 12.985, 32.484 and 49.939, respectively. The P values are 9%, 0.2% and 0.1%, which are less than 10% or 5%. Although economically restrained environmental regulation has a significant impact on haze pollution, it increases haze pollution. The reason for this finding is that the purpose of enterprise production is to gain profits. As long as the sewage fee is less than the profit, the enterprise will continue to produce. When a government department levies sewage fees to sewage disposal enterprises, the enterprise believes that it has paid economic compensation for its sewage discharge actions, and they may continue to discharge pollutants to the environment. The collection of sewage fees has caused enterprises to lose some economic benefits. Enterprises will recoup their losses by increasing production. At this time, more pollutants may be produced, which will cause more pollution to the environment.

The influence coefficient of public participation in environmental regulation on haze pollution changes from positive to negative. In the low quantile, the marginal effect of public participation in environmental regulation on haze pollution is 0.121. The P value is 8.5%, which is less than 10%. Public participation in environmental regulation has a significant impact on

haze pollution, but the result is that the more public participation there is, the higher the haze pollution. In the high quantile, the marginal effect of public participation in environmental regulation on haze pollution is -0.002. The P value is 99%, which is greater than 10%. Public participation in environmental regulation may reduce haze pollution, but its effect is insignificant. In China, although public participation in environmental protection is increasingly strengthened, most of the public believe that their personal strength is too small, and the effect on environmental protection is minimal. Over time, the public will adopt an indifferent attitude toward environmental pollution cases [66]. When faced with pollution, people will defer to the actions of countries and governments. Even if they stand up, they will not affect the results. Therefore, the public will adopt a negative attitude toward participation in environmental protection [67, 68].

Among the other control variables, the coefficient of industrial concentration changes from a negative value to a positive value along the quantile levels. In the low quantiles, the marginal effect of the industrial concentration on haze pollution is -0.158. The P value is 81.1%, which is greater than 5%. The industrial concentration has no significant effect at low haze pollution levels. In the high quantiles, the industrial concentration has a significant impact on haze pollution. For every 1 unit increase in industrial concentration, haze pollution increased by 0.655, 1.235 and 1.753 units, which means that China's industries are focused on making profits and taking less social responsibility. The marginal effect of enterprise scale on haze pollution changes from -1.193 to -1.850, and the absolute value increases. This shows that the larger the enterprise scale, the more capable it is to control haze pollution. The influence coefficient of the science and technology input intensity on haze pollution changes from -0.498 to -1.093. Regardless of the quantile level, science and technology input intensity can reduce haze pollution. The P values were 78.9%, 64.3%, 49.4%, 53.7% and 63.0%. These P values are greater than 5%, indicating that the effect of science and technology input intensity on haze pollution is not significant. This means that the ultimate goal of China's science and technology investment is still to achieve profitable "red" technological innovation and that relatively little is invested in "green" technological innovation for environmental protection.

The variation in the regression coefficient of each explanatory variable by quantile, based on further analysis, is shown more intuitively in Fig 2. For example, the panel in the first column of the first row shows that as the quantile changes, there is a downward trend in the quantile regression coefficient of the impact of command-and-control environmental regulation. If command-and-control environmental regulation can play a significant role, haze pollution in China will decline. The basic shape of this panel confirms the pattern of the decline in the quantile regression coefficient of the command-and-control environmental regulation in Table 1. On the other hand, Fig 1 also shows that at both ends of the conditional distribution, the 95% confidence interval becomes wider, and the standard error of the regression coefficient estimate is larger, which makes the estimation of the quantile regression coefficient at both ends inaccurate.

## Robustness test

To assess the robustness of the results, the following tests were carried out in this research. Due to the time lag of environmental regulation, previous environmental policies may have an impact on the current haze pollution. Therefore, the lag term of environmental regulation is used as an explanatory variable for the panel quantile regression. We randomly selected half of the data from the total sample for testing. The test results are not shown due to space limitations. The test results are basically consistent with the previous research results, and the research conclusions in this paper are consistent and reliable.

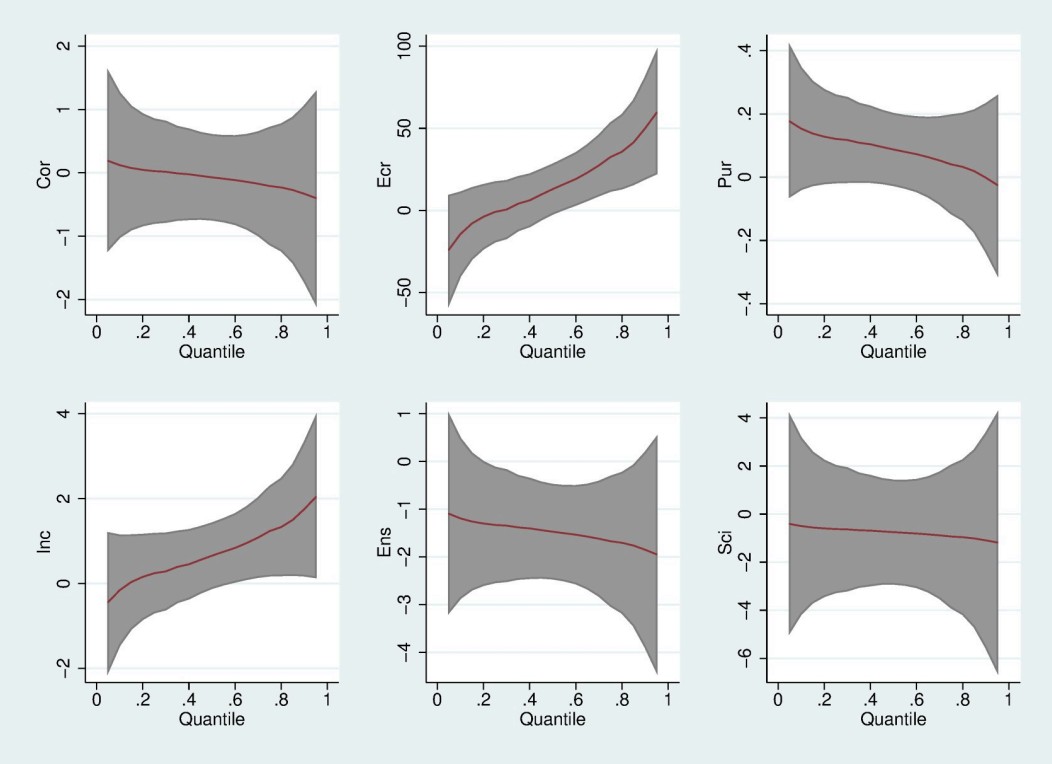

**Fig 2. Variation in the panel quantile regression coefficients.**

## Conclusion

### Research conclusion

Command-and-control environmental regulation can reduce haze pollution in low to high quantiles, but its impact is not significant. This is mainly because the main body of environmental law enforcement is divided, the powers and responsibilities are scattered, and departmental coordination is difficult. The shocking role of the command-and-control environmental regulation in China has not yet played out. The Chinese government should attach great importance to this situation. In the low quantiles, economically restrained environmental regulation can reduce the level of haze pollution. However, in the high quantiles, economically restrained environmental regulation cannot reduce the level of haze pollution. The collection of sewage fees is far less than the total profit; therefore, these fees do not have a deterrent effect on enterprises. Although public participation in environmental regulation can reduce the level of haze pollution in high quantiles, its impact becomes insignificant over time. The reason for this trend is that the public's intuitive perception of mild to moderate air pollution is not profound [69]. When haze is severe, the lives and travel plans of the public will be severely affected, and their willingness to control haze will rise. The governmental departments responsible for environmental management need to properly guide the public to fight against pollution and make the public responsible for third-party supervision of environmental pollution control [70].

### The countermeasures

Based on the abovementioned empirical results and combined with China's actual situation, this paper proposes countermeasures and suggestions to address haze pollution.

Enhancing environmental law enforcement. At present, there are two main reasons for the weak enforcement of environmental law. The first is the management system based on the administrative district, and the local protectionism is serious; The second is that the main body of environmental law enforcement is divided, the powers and responsibilities are scattered, the department coordination is difficult, and the law enforcement costs are high. It is proposed to divide the country into several regions and set up regional management agencies to serve as the dispatching agencies of the national environmental protection administrative departments. Drawing on the special commissioner system for tax auditing, those involved in the environmental auditing special dispatcher system are responsible for supervising the implementation of local environmental protection responsibilities. To ensure that the inspectors perform their duties, they shall be appointed by state staff at or above the deputy ministerial level and the state council and will be held accountable to the state council.

Increasing the amount of the sewage fees collected. The fees collected for pollutant discharge are far less than the total profit. The state should greatly increase the standard for the collection of sewage fees, forcing enterprises to incorporate the cost of pollutant discharge into the cost of products, and transforming the cost of energy saving, emission reduction and environmental pollution control into an inherent requirement of industrial enterprises to reduce production costs. For enterprises to obtain more benefits, environmental costs must become one of the primary considerations. Reducing environmental pollution will be the only way for enterprises to survive.

Increasing interactive public participation platforms. The use of public participation platforms in western countries has shown that without the support and participation of residents, the state's administrative management and the self-discipline of enterprises to control environmental pollution will be overwhelming and lead to unpredictable results. The state, enterprises, and residents should become the three main bodies of environmental protection. Relying solely on the state and enterprises to control environmental pollution, without the active participation of residents, environmental protection will lead to a loss of important support. An interactive public participation platform should be created to improve the enthusiasm and effectiveness of public participation.

This paper analyzes these reasons for this finding and gives countermeasures. The reasons for this result are complex, and some analyses were performed in this research. Because haze pollution has no boundaries, the haze status in one region will affect that in the others. In future research, we will consider the interaction between different regions and use a spatial panel quantile model.

## Supporting information

**S1 Data.**
(RAR)

## Author Contributions

**Methodology:** Guozhu Li.

**Writing – original draft:** Congxin Li.

**Writing – review & editing:** Guozhu Li.

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
