## [Editor Report · Decision Letter 0]

8 Jun 2020

PONE-D-20-14570

Dose Environmental Regulation Reduce China's Haze Pollution?

Empirical Analysis Based on Panel Quantile Regression

PLOS ONE

Dear Dr. Li,

Thank you for submitting your manuscript to PLOS ONE. After careful consideration, we feel that it has merit but does not fully meet PLOS ONE’s publication criteria as it currently stands. Therefore, we invite you to submit a revised version of the manuscript that addresses the points raised during the review process.

We look forward to receiving your revised manuscript.

Kind regards,

Bing Xue, Ph.D.

Academic Editor

PLOS ONE

Journal Requirements:

Additional Editor Comments (if provided):

(1) most of the references are from Chinese journals, I'd like to suggest to include more English references in your manuscript.

(2) The thesis did a relatively complete review, but there was a lack of discussion in the results section. Therefore, the author needs to add a discussion section to discuss the analysis results and make recommendations, and finally draw out the conclusion.
---

## [Author Response · Author response to Decision Letter 0]

18 Jun 2020

Dear Editors and Reviews of PLOS ONE：

 We really appreciate your consideration and all the excellent suggestions for improving this paper. Taking all the valuable comments into consideration, we have revised the paper, which is entitled “Does environmental regulation reduce China's haze pollution? empirical analysis based on panel quantile regression” (PONE-D-20-14570).We hope the revision meets the standards of PLOS ONE. In addition, we provided a point-by-point response to each of your comment.

Journal Requirements:

Response:

 We have revised the article with reference to the relevant content.

Response:

 Thanks for your helpful feedback, we have modified it as required.

Response:

 According to your comment, we have updated my Information and clicked on the Fetch/Validate link next to the ORCID field.

4. Please include captions for your Supporting Information files at the end of your manuscript, and update any in-text citations to match accordingly.

Response:

 We have put all the information in the main content of the paper.

Additional Editor Comments (if provided):

(1) most of the references are from Chinese journals, I'd like to suggest to include more English references in your manuscript.

Response:

 Thanks for your helpful comment, we have cited more English references and deleted some Chinese journals. The blue is the deleted Chinese journals and the red is the added English references.

Xu D., Huang Z.F., Huang R.,2019. The spatial effects of haze on tourism flows of Chinese cities: Empirical research based on the spatial panel econometric model. Acta Geographica Sinica. 74(4):814-830. DOI: 10.11821/dlxb201904014

Shao S., Li X., Cao J.H., Yang L.L.,2016. China's Economic Policy Choices for Governing Smog Pollution Based on Spatial Spillover Effects. Economic Research Journal. 9,73-88.

Liu HJ, Lei MY. The dilemma of the regional collaborative governance of haze pollution and its solution ideas in China. China Population, Resources and Environment. 2018;28(10):88-95. doi: 10.12062 /cpre.20180507.

Li C.X., Li G.Z.,2017. The Study on Innovation Performance Evaluation of High Energy-consuming Industries in Hebei Province. Mathematics in Practice and Theory. 47(5):55-62. 

Lan QJ, Chen CF. Soft Institution，Public Recognition and Efficiency of Air Pollution Abatement. China Population, Resources and Environment.2015;25(9):145-152

 doi: 10. 3969 /j. issn. 1002 －2104. 2015. 09. 019

Li W.D., Huang X.,2018. Emprical Study on the Social and Economic Influence

 Factors of Beijing’s Haze. Journal of Capital University of Economics and Business. 20(4):58-67. DOI:10. 13504/j.cnki.issn1008-2700.2018.04.007.

Yu G.Y., Xiu C.L.,2018. Effects of Urbanization to Haze Pollution in Liaoning Province Based on Spatial Perspective. Economic Geography. 38(4):100-108,122. DOI:10.15957/j.cnki.jjdl.2018.04.012.

Cheng Z.H., Liu J., Li L.S.,2019.Research on the Effects of Industrial Structure Adjustment and Technical Progress on Haze Reduction. China Soft Science. 1,146-154.

Wei W.X., Ma X.L.,2015.Optimal Policy for Energy Structure Adjustment and Haze Governance in China. China Population, Resources and Environment. 25(7):6 -14. DOI: 10. 3969 /j. issn. 1002-2104.2015.07.002.

Zhang X.B., Wang J.Z.,2019.The spatial effect of region energy efficiency on haze pollution—Empirical analysis based on the Spatial Durbin Model. China Environmental Science.39(4):1371-1379. DOI:10.19674/j.cnki.issn1000-6923.2019.0165.

Yan Y.X., Qi S.Z.,2017. FDI and Haze Pollution in China. Statistical Research.35(4):69-81. DOI: 10 19343 /j cnki 11- 1302 /c 201705007.

Yan Y.X., Qi S.Z.,2017.Time-space effect test on foreign direct investment and PM2.5

 pollution at city level．China Population, Resources and Environment.27(4):68-77. DOI:10.12062/cpre.20170321.

Chen RJ, Kan HD. Haze/fog and human health: A literature review. Chinese Journal of Nature. 2014;35(5):342-344. doi:10.3969/j.issn.0253-9608.2013.05.006. 

Yan Y.B., Zhang J.,2016.Impact of smog weather on the amount of inbound tourists of China based on the natural trend curve. Economic Geography. 36(12): 183-188.

 DOI:10.15957/j.cnki.jjdl.2016.12.026.

Xu D., Huang Z.F., Huang R., Hou G.L., Cao F.D.,2019. The spatiotemporal dynamic correlation analysis of haze pollution and inbound tourism in central and eastern China. Journal of Natural Resources. 34(5): 1108-1120. DOI: 10.31497/zrzyxb.20190516.

Yi L., Zhou Y.N., Li C.P., Yang L.,2018. Analysis of the effects of driving restriction policies in controlling haze pollution. China Population, Resources and Environment. 28( 10) : 81-87． DOI: 10. 12062 /cpre. 20180509.

Zhang L.W., Cheng D.P., Xu L.L.,2019. An Analysis of the Effect of Environmental Protection Policies on Smog Control in the New Era: Based on the Perspective of the PM2.5 Concentration Change. Journal of Shanghai University of Finance and Economics. 21(2):17-28. DOI: 10.16538/j.cnki.jsufe.2019.02.002.

3.Sun JK, Zhang JH, Wang C, Duan XF. Escape or stay? Effects of haze pollution on domestic travel: Comparative analysis of different regions in China.  Science of The Total Environment.2019; 690:151-157. doi:10.1016/j.scitotenv.2019.06.415

4.Xu XW, Tong DX, Wang YL, Wang SY. The Impacts of Different Air Pollutants on Domestic and Inbound Tourism in China. International Journal of Environmental Research and Public Health .2019;16:5127,1-15 doi:10.3390/ijerph16245127

5.Korhonen J, Pätäri S, Toppinen; A, Tuppura A. The role of environmental regulation in the future competitiveness of the pulp and paper industry: the case of the sulfur emissions directive in Northern Europe. Journal of Cleaner Production. 2015;108:864-872. doi:10.1016/j.jclepro.2015.06.003

7.Liu XH, Xia HX. Empirical analysis of the influential factors of haze pollution in china—Based on spatial econometric model. Energy & Environment . 2018;30(5):1-13 doi:10.1177/0958305X18813648

8.Chen JX, Zhang YG, Zheng SL. Ecoefficiency, environmental regulation opportunity costs, and interregional industrial transfers: Evidence from the Yangtze River Economic Belt in China. Journal of Cleaner Production.2019;233:611-625.

doi.org/10.1016/j.jclepro.2019.06.117

9.Huang FF, Li X, Gao XH. PM2.5 Spatiotemporal Variations and the Relationship with Meteorological Factors during 2013-2014 in Beijing, China. PloS One. 2015;10(11):1-33. doi: 10.1371/journal.pone.0141642

12.Wang X, Lei P. Does strict environmental regulation lead to incentive contradiction? -Evidence from China. Journal of Environmental Management.2020;269:1-12

 doi:10.1016/j.jenvman.2020.110632

14.Zhou Q, Zhang XL, Shao QL, Wang XL. The non-linear effect of environmental regulation on haze pollution: Empirical evidence for 277 Chinese cities during 2002–2010.Journal of Environmental Management.2019;248:109274 . doi.org/10.1016/j.jenvman.2019.109274

20.Zhang MW, Fang GX, Yin WH, Xie B, Ren MZ, Xu ZC, et al.Airborne PCDD/Fs in two e-waste recycling regions after stricter environmental regulations. Journal of Environmental Sciences. 2017;62:3-10. doi:10.1016/j.jes.2017.07.009

21.Borsatto JMLS, Amui LBL. Green innovation: Unfolding the relation with environmental regulations and competitiveness. Resources, Conservation & Recycling.2019;149:445-454. doi.org/10.1016/j.resconrec.2019.06.005

24.Cao K, Zhang WT, Liu SB, Huang B, Huang W. Pareto law-based regional inequality analysis of PM2.5 air pollution and economic development in China.Journal of Environmental Management.2019;252:1-9

doi.org/10.1016/j.jenvman.2019.109635.

27.Gan T, Liang W, Yang HC, Liao XC. The effect of Economic Development on haze pollution (PM2.5) based on a spatial perspective: Urbanization as a mediating variable. Journal of Cleaner Production.2020;266:1-14. doi.org/10.1016/j.jclepro.2020.121880

30.Khan Z, Shahbaz M, Ahmad M, Rabbi F, Yang SQ.Total retail goods consumption, industry structure, urban population growth and pollution intensity: an application of panel data analysis for China. Environmental Science and Pollution Research.2019;36(21): 32224-32242. doi: 10.1007/s11356-019-06326-0

31.Zhao XF, Jian H, Wang HN, Zhao JJ, Qiu QY, Tapper N, et al. Remotely sensed thermal pollution and its relationship with energy consumption and industry in a rapidly urbanizing Chinese city. Energy Policy.2013;57:398-406. doi.org/10.1016/j.enpol.2013.02.007

33.He LY, Zhang L, Liu RY.Energy consumption, air quality, and air pollution spatial spillover effects: evidence from the Yangtze River Delta of China.Chinese Journal of Population Resources and Environment.2019;17(4):329-340.

 doi: 10.1080/10042857.2019.1650245

35.Zheng YM.Effect of FDI on China's environmental pollution: Evidence based on spatial panel data. Ecological Economy.2018;2:141-146.

 doi:CNKI:SUN:STJY.0.2018-02-010

36.Hille E, Shahbaz M, Moosa I.The impact of FDI on regional air pollution in the Republic of Korea: A way ahead to achieve the green growth strategy?. Energy Economics. 2019;81:308-326. doi:10.1016/j.eneco.2019.04.004

37.Ponce P, Alvarado R. Air pollution, output, FDI, trade openness, and urbanization: evidence using DOLS and PDOLS cointegration techniques and causality. Environmental Science and Pollution Research. 2019;26 (19):19843-19858. 

 doi:10.1007/s11356-019-05405-6

40.Martinelli N, Girelli D, Cigolini D, Sandri M, Ricci G, et al. Access rate to the emergency department for venous thromboembolism in relationship with coarse and fine particulate matter air pollution.[J].PloS one,2012,7(4):e34831.

 doi:10.1371/journal.pone.0034831

41.Xiang H, Mertz KJ, Arena VC, Brink LL, Xu X, Bi Y, Talbott EO. Estimation of short-term effects of air pollution on stroke hospital admissions in Wuhan, China. PLoS One. 2013 Apr 12;8(4):e61168. doi: 10.1371/journal.pone.0061168. 

42.Rajper SA, Ullah S, Li Z. Exposure to air pollution and self-reported effects on Chinese students: A case study of 13 megacities.PloS One.2018;13(3):e0194364

 doi: 10.1371/journal.pone.0194364

45.Deng TT, Li X, Ma ML.Evaluating impact of air pollution on China’s inbound tourism industry: a spatial econometric approach. Asia Pacific Journal of Tourism Research.2017;22(7):771-780. doi.org/10.1080/10941665.2017.1331923

46.Zhou XG, Jiménez YS, Pérez Rodríguez JV, Hernández JM. Air pollution and tourism demand: A case study of Beijing, China. International Journal of Tourism Research. 2019;747-757. doi:10.1002/jtr.2301

47.Cook BJ.The Politics of Market-Based Environmental Regulation: Continuity and Change in Air Pollution Control Policy Conflict. Social Science Quarterly. 2002;83(1):156-166. doi:10.1111/1540-6237.00076

49.Jia SW, Liu XL, Yan GL. Effect of APCF policy on the haze pollution in China: A system dynamics approach. Energy Policy.2019;125:33-44. doi:10.1016/j.enpol.2018.10.012

50.Wu XP, Gao M, Guo SH, Li W. Effects of environmental regulation on air pollution control in China: a spatial Durbin econometric analysis. Journal of Regulatory Economics. 2019; 55 (3):307-333. doi:10.1007/s11149-019-09384-x

(2) The thesis did a relatively complete review, but there was a lack of discussion in the results section. Therefore, the author needs to add a discussion section to discuss the analysis results and make recommendations, and finally draw out the conclusion.

Response:

 Following your suggestion, we have modified this part of the content. The relevant content is shown in article.

Thanks again

Guozhu Li

2020.06.17

---

## [Decision Letter · Decision Letter 1]

27 Aug 2020

PONE-D-20-14570R1

Dose Environmental Regulation Reduce China's Haze Pollution?

Empirical Analysis Based on Panel Quantile Regression

PLOS ONE

Dear Dr. Li,

Thank you for submitting your manuscript to PLOS ONE. After careful consideration, we feel that it has merit but does not fully meet PLOS ONE’s publication criteria as it currently stands. Therefore, we invite you to submit a revised version of the manuscript that addresses the points raised during the review process.

We look forward to receiving your revised manuscript.

Kind regards,

Bing Xue, Ph.D.

Academic Editor

PLOS ONE

Reviewers' comments:

Reviewer's Responses to Questions

**Comments to the Author**

1. If the authors have adequately addressed your comments raised in a previous round of review and you feel that this manuscript is now acceptable for publication, you may indicate that here to bypass the “Comments to the Author” section, enter your conflict of interest statement in the “Confidential to Editor” section, and submit your "Accept" recommendation.

Reviewer #1: All comments have been addressed

Reviewer #2: All comments have been addressed

2. Is the manuscript technically sound, and do the data support the conclusions?

Reviewer #1: Yes

Reviewer #2: Yes

3. Has the statistical analysis been performed appropriately and rigorously? 

Reviewer #1: Yes

Reviewer #2: Yes

4. Have the authors made all data underlying the findings in their manuscript fully available?

Reviewer #1: Yes

Reviewer #2: Yes

5. Is the manuscript presented in an intelligible fashion and written in standard English?

Reviewer #1: No

Reviewer #2: No

6. Review Comments to the Author

Reviewer #1: Few minor changes are required before final placement:

Abstract can be polished more considering scientific vigor.

English style checks and typos need to be looked after while finalizing.

Try to reduce some information in the methodology section.

Conclusion section is under par. Revise it and improve it considering important findings and policy implications.

Reviewer #2: Although efforts have been made to improve its English, there are still several grammatical errors in the manuscript that the authors need to give attention to so as to make the paper fit the standard English requirements.

7. PLOS authors have the option to publish the peer review history of their article (what does this mean?). If published, this will include your full peer review and any attached files.

Reviewer #1: **Yes: **Ghaffar Ali

Reviewer #2: No

---

## [Author Response · Author response to Decision Letter 1]

12 Sep 2020

Dear Editors and Reviewers：

 First of all, thank you very much for your very encouraging and inspiring feedback on my work and for your very constructive and helpful comments that have definitely improved the paper a great deal. Taking all the valuable comments into consideration, we have revised the paper, which is entitled “Does environmental regulation reduce China's haze pollution? an empirical analysis based on panel quantile regression” (PONE-D-20-14570R1).We hope the revision meets the standards of PLOS ONE. In addition, we provided a point-by-point response to each of your comments.

Response to Reviewer 1’s Comments

1.Abstract can be polished more considering scientific vigor.

Response:

 Thanks for your insightful comment. According to your comment, we have polished the abstract more considering scientific vigor. The abstract introduces the background, research methods and conclusions. In the conclusion, we have analyzed the degree and direction of the influence of different environmental regulations. We also added the reasons for different policy effects. The relevant content is shown as follows.

 Haze pollution in China is very serious and has become the source of mortality, affecting the health and lives of residents. The Chinese government adopts different policy measures to reduce haze pollution. The impact of different types of environmental regulations on haze pollution has become a hot topic for academics and government departments. Based on panel data from 2005-2017, this paper studies the effect of different types of environmental regulations on haze pollution in 30 provinces of China using a panel quantile model. The results show that when haze pollution changes from a low quantile to a high quantile, the marginal impact of command-and-control environmental regulation on haze pollution changes from 0.122 to -0.332. Command-and-control environmental regulation can reduce haze pollution, but its impact is not significant. This result shows that environmental law enforcement is not strict. The marginal impact of economically restrained environmental regulation on haze pollution changes from -14.389 to 49.939. Economically restrained environmental regulation can reduce haze pollution in low quantiles, but not in high quantiles. The collection of sewage charges fees is far less than the total profit , which has no deterrent effect on enterprises. The marginal impact of public participation in environmental regulation on haze pollution changes from 0.154 to -0.002. Public participation in environmental regulation cannot reduce haze pollution in low quantiles , but can in high quantiles, however its impact becomes insignificant. In China, most of the public believe that their effect on environmental protection is minimal. This study reveals the quantile-based discrepancy in the effect of environmental regulation on haze pollution, and offers a new perspective for research on the effects of environmental regulation.

2.English style checks and typos need to be looked after while finalizing.

Response:

 Thanks deeply for your suggestion. We read the whole article carefully and revised the grammar and typos. Furthermore, in this revision, grammatical and writing style errors in the original version have been refined and edited by our friend who is a native English speaker. The red part of the article is the content added or modified.

3. Try to reduce some information in the methodology section.

Response:

 We appreciate your suggestion for this issue. Lines 136 to 141 in revised Manuscript with Track Changes are the simplified process of formula 2. According to your comment, we deleted these. The blue is the deleted content.

 The test function is defined as

 (3)

 In formula 3, is a demonstrative function and satisfies the following,

 (4)

 Therefore, formula 2 can be expressed equivalently as follows,

 (5)

4. Conclusion section is under par. Revise it and improve it considering important findings and policy implications.

Response:

 Thanks for your helpful feedback. The last part of the manuscript is Discussion. Discussion includes "Research results, The countermeasures and Conclusion". Conclusion is a simple summary of the research results. There is not much analysis of policy effects. If the conclusion is the last section, it is indeed a bit too single. According to your comment, we have adjusted the last part. Conclusion includes "Research Conclusion and The countermeasures". We removed the original conclusion section and turned it into the last paragraph. The last paragraph becomes the future research of this study.

Response to Reviewer 2’s Comments

Although efforts have been made to improve its English, there are still several grammatical errors in the manuscript that the authors need to give attention to so as to make the paper fit the standard English requirements.

Response:

 Thanks deeply for your suggestion. We read the whole article carefully and revised the grammar and typos. Furthermore, in this revision, grammatical and writing style errors in the original version have been refined and edited by our friend who is a native English speaker. The red part of the article is the content added or modified.

---

## [Decision Letter · Decision Letter 2]

2 Oct 2020

Does environmental regulation reduce China's haze pollution?an empirical analysis based on panel quantile regression

PONE-D-20-14570R2

Dear Dr. Li,

We’re pleased to inform you that your manuscript has been judged scientifically suitable for publication and will be formally accepted for publication once it meets all outstanding technical requirements.

Kind regards,

Bing Xue, Ph.D.

Academic Editor

PLOS ONE

Additional Editor Comments (optional):

Reviewers' comments:

Reviewer's Responses to Questions

**Comments to the Author**

1. If the authors have adequately addressed your comments raised in a previous round of review and you feel that this manuscript is now acceptable for publication, you may indicate that here to bypass the “Comments to the Author” section, enter your conflict of interest statement in the “Confidential to Editor” section, and submit your "Accept" recommendation.

Reviewer #2: All comments have been addressed

2. Is the manuscript technically sound, and do the data support the conclusions?

Reviewer #2: Yes

3. Has the statistical analysis been performed appropriately and rigorously? 

Reviewer #2: Yes

4. Have the authors made all data underlying the findings in their manuscript fully available?

Reviewer #2: Yes

5. Is the manuscript presented in an intelligible fashion and written in standard English?

Reviewer #2: Yes

6. Review Comments to the Author

Reviewer #2: All comments have been addressed including English language minor errors. Therefore, the manuscript could be accepted.

7. PLOS authors have the option to publish the peer review history of their article (what does this mean?). If published, this will include your full peer review and any attached files.

Reviewer #2: **Yes: **Dr Essiagnon John-Philippe Alavo

---

## [Editor Report · Acceptance letter]

7 Oct 2020

PONE-D-20-14570R2 

Does environmental regulation reduce China's haze pollution?an empirical analysis based on panel quantile regression 

Dear Dr. Li:

I'm pleased to inform you that your manuscript has been deemed suitable for publication in PLOS ONE. Congratulations! Your manuscript is now with our production department. 

Kind regards, 

on behalf of

Professor Bing Xue 

Academic Editor

PLOS ONE